# Does Carrier Envelope Phase Affect the Ionization Site in a Neutral Diatomic Molecule?

Alex Schimmoller [†], Harrison Pasquinilli [†] and Alexandra S. Landsman *

Department of Physics, The Ohio State University, Columbus, OH 43210, USA; schimmoller.11@osu.edu (A.S.)
* Correspondence: landsman.7@osu.edu
† These authors contributed equally to this work.

**Abstract:** A recent work shows how to extract the ionization site of a neutral diatomic molecule by comparing Quantum Trajectory Monte Carlo (QTMC) simulations with experimental measurements of the final electron momenta distribution. This method was applied to an experiment using a 40-femtosecond infrared pulse, finding that a downfield atom is roughly twice as likely to be ionized as an upfield atom in a neutral nitrogen molecule. However, an open question remains as to whether an assumption of the zero carrier envelope phase (CEP) used in the above work is still valid for short, few-cycle pulses where the CEP can play a large role. Given experimentalists' limited control over the CEP and its dramatic effect on electron momenta after ionization, it is desirable to see what influence the CEP may have in determining the ionization site. In this paper, we employ QTMC techniques to simulate strong-field ionization and electron propagation from neutral $N_2$ using an intense 6-cycle laser pulse with various CEP values. Comparing simulated electron momenta to experimental data indicates that the ratio of down-to-upfield ions remains roughly 2:1 regardless of the CEP. This confirms that the ionization site of a neutral molecule is determined predominantly by the laser frequency and intensity, as well as the ground-state molecular wavefunction, and is largely independent of the CEP.

**Keywords:** strong field ionization; molecular ionization; Quantum Trajectory Monte Carlo





## 1. Introduction

Tunneling occurs when a laser's strong electric field distorts the Coulombic barrier of an atom enough to allow for the electron to escape [1]. For the case of a diatomic molecule, this picture is complicated by the presence of a double-well potential. This leads to two possible ionization sites, as shown in Figure 1: the upfield (higher energy) atom and the downfield (lower energy) atom [2]. Commonly used theories of molecular ionization, such as molecular ADK [3], molecular SFA, and the partial Fourier transform approach [4], assume implicitly that all ionization is downfield, corresponding to the bound electron wavepacket adiabatically responding to the relatively low-frequency laser field. However, it is known that ionization in charged molecules can occur from either atom depending on the internuclear separation, alignment, and other conditions [2].

When a positively charged diatomic molecule begins to dissociate, the resulting bond softening traps the electron in the upper well and leads to upfield ionization. This process is known as ionization enhancement [5–10] and has been repeatedly confirmed in experiments that examine molecular fragments following a Coulomb explosion [7,8,11,12]. However, until recently, there was no technique for determining the ionization site in neutral atoms. A recent work suggests that the longitudinal photoelectron momentum distributions for charged ions could be looked at to identify the ionization location [13]. The principle behind this technique is that the electron experiences different forces due to the Coulomb potential depending on which atom it is ionized from. If it is ionized from the downfield atom, it will propagate directly into the continuum, but if it is from the upfield atom, it will first

have to pass the downfield atom, distorting its trajectory compared to the case of ionization from an atom with the same binding potential. Additionally, if the electron tunnels from the upfield atom to the downfield atom, there will be a delay in ionization, causing a shift in the photoelectron momentum distribution (PMD) for circularly or elliptically polarized light. However, the approach presented in [13] views the ionization process as either all upfield or all downfield, and therefore does not provide a method for quantifying upfield to downfield ionization when both contributions are significant. A more recent work by Ortmann and colleagues [14] presents a method for quantifying the ratio of upfield to downfield ionization events, finding a significant contribution from both under typical experimental conditions that employ infrared light for strong field ionization.

In this work, we focus on the approach presented in [14], which establishes a quantitative procedure for finding the ionization site in a neutral diatomic molecule. This procedure relies on simulating a variety of upfield:downfield ionization site ratios and determining which ratio matches the experimental momentum distribution. We expand upon this technique by examining the effect that the carrier envelope phase (CEP) has upon the results to see if the approach requires stabalizing the CEP or if it can work over a random CEP distribution.

It is important to check the robustness of this model with respect to changing the carrier envelope phase for two reasons. First, it expands the range of applicability of the approach in [14] to few-cycle pulses without requiring CEP averaging or pulse stabilization, which would introduce additional sources of uncertainty. Changing the CEP can change the final PMD, which may affect the technique as it depends on comparing the final transverse momenta of the electrons in order to determine the ionization site. This is not an issue for longer pulses where PMDs are independent of CEP, but it can play an important role for few-cycle pulses where the CEP is not stabilized. Stabilizing the CEP is a non-trivial task experimentally, let alone setting it to a specific value [15]. Second, the robustness of the ionization site calculation to CEP changes supports the view that upfield ionization is a non-adiabatic effect determined by the Keldysh parameter, $\gamma = \omega \sqrt{2I_p}/E_0$, which is independent of the CEP. Here, $\omega$, $E_0$, and $I_p$ are the laser frequency, peak field strength, and ionization potential, respectively (atomic units are assumed throughout this text).

The remainder of our work is organized as follows. Section 2 describes the techniques and simulation used to calculate the PMDs. Section 3 analyzes the results, finding that CEP does not have a significant impact on the relative contribution of upfield to downfield ionization. Section 4 concludes and summarizes.

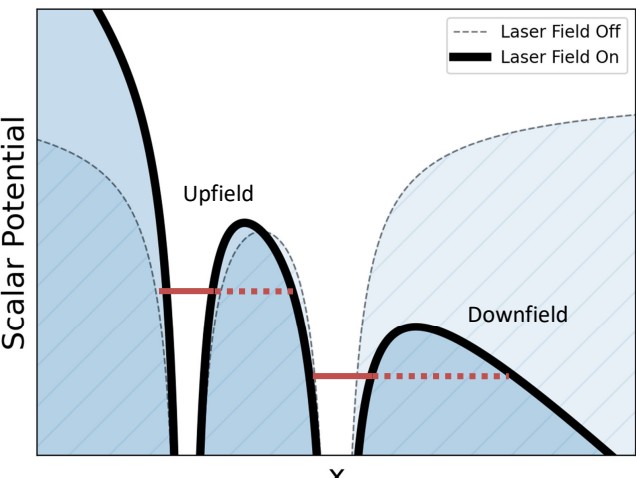

**Figure 1.** Schematic of the electric potential created by a diatomic molecule with and without a strong laser field present. Turning on the laser electric field allows electrons originating from both atomic sites to tunnel into the continuum. Upfield and downfield electrons experience different potentials due to the molecule's asymmetric Coulomb forces, altering their trajectories.

## 2. Simulating $N_2$ Strong Field Ionization

To highlight the role that the carrier envelope phase can play in the final momentum distribution, consider a laser pulse linearly polarized along the x-axis with electric field profile

$$\mathbf{E}(t) = E_0 \cos(\omega t + \phi)\text{env}(t)\hat{\mathbf{x}}, \tag{1}$$

where $\omega = 2\pi c/\lambda$ is the angular frequency of the wave, $\text{env}(t) = \cos^2(\frac{\omega t}{2N})$ is an envelope centered at $t = 0$ containing $N$ laser cycles, and $\phi$ is the carrier envelope phase. According to the strong field approximation [16], ionization is most likely near the absolute maxima of the laser pulse and the electron's final momentum is largely determined by the vector potential at the time of ionization $\mathbf{A}(t_0) = -\int^{t_0} \mathbf{E}(t')dt'$. Therefore, the CEP influences both the ionization time and final electron momenta. When ionizing diatomic molecules [14], it also controls when parent atoms are considered either upfield or downfield. For long pulses, this poses no problem since the field and vector potential can be approximated as plane waves and ionization takes place over many optical cycles. However, for short pulses when $N$ is on the order of a few optical cycles and the envelope function $\text{env}(t)$ decays quickly away from $t = 0$, only the centermost field peaks contribute to ionization, amplifying the CEP's influence on the photoelectron distribution.

The setup for the simulation closely follows that of reference [14]: a six-cycle, linearly polarized laser pulse of the form (1) with wavelength $\lambda = 800$ nm and peak intensity $I_0 = 1.3 \cdot 10^{14}$ W/cm$^2$ is incident upon a neutral $N_2$ molecule with ionization potential $I_p = 15.6$ eV. Focal averaging is applied to the intensity profile by assigning to each intensity $I$ a relative weight $\sim \frac{2I + I_0}{I^{5/2}}\sqrt{I_0 - I}$ [17–19]. In the simulation, intensities are sampled according to these weights and used to determine the peak electric field $E_0 = \sqrt{I}$ for subsets of the simulated electrons. The molecule is tilted $\theta = 45$ degrees against the polarization direction with nitrogen atoms located at positions $\mathbf{r}_A = \frac{R_0}{2\sqrt{2}}(-1, 0, -1)$ and $\mathbf{r}_B = \frac{R_0}{2\sqrt{2}}(1, 0, 1)$ a.u., respectively, where $R_0 = 2$ a.u. is the internuclear distance. This tilt creates an asymmetric Coulomb force acting on electrons originating from each of the parent nuclei. When the laser pulse is incident upon the molecule, electrons may be ionized from either the up- or downfield atom (this designation alternates depending on whether the electric field is positive or negative). They then propagate semiclassically until the end of the laser pulse, when their positions and momenta are recorded and asymptotic momenta are calculated.

Initial conditions for ionized electrons are achieved via Monte Carlo reject sampling [20,21]. With the choice of the up- or downfield parent atom fixed, ionization times $t_0$ and initial transverse velocities $v_\perp = \sqrt{v_{0,y}^2 + v_{0,z}^2}$ are fed into a reject-sampling algorithm that compares the (normalized) ionization rate to randomly generated values. This ionization rate accounts for the molecular orbital by importing the electronic wavefunction in $N_2$ from GAMESS [22] and performing a partial Fourier transform to obtain the electron's initial transverse velocity distribution. Electrons tunnel nonadiabatically to the continuum according to the ionization theory presented in reference [23], though with the more general field profile (1) containing both the enveloping function and carrier envelope phase. The atomic ionization rate $W(t_0, v_\perp)$ for an electron ionized at time $t_0$ and with transverse velocity $v_\perp$ is

$$W(t_0, v_\perp) = \frac{\omega^2(2I_p)^{5/2}}{2[E_0\text{env}(t_0)]^4\gamma^2(t_0, v_\perp)[\gamma^2(t_0, v_\perp) + \cos^2(\omega t_0 + \phi)]\cos^2(\omega t_0 + \phi)}$$

$$\times \exp\left(-\frac{[E_0\text{env}(t_0)]^2}{\omega^3}\left\{\left[\sin^2(\omega t_0 + \phi) + \gamma^2(t_0, v_\perp) + \frac{1}{2}\right] \times \sinh^{-1}\gamma(t_0, v_\perp)\right.\right. \tag{2}$$

$$\left.\left. -\frac{1}{2}\gamma(t_0, v_\perp)\sqrt{1 + \gamma^2(t_0, v_\perp)}(1 + 2\sin^2(\omega t_0 + \phi))\right\}\right),$$

where

$$\gamma(t_0, v_\perp) = \omega \frac{\sqrt{2I_p + v_\perp^2}}{|\mathbf{E}(t_0)|} \tag{3}$$

is the effective Keldysh parameter [23]. Electrons with ionization times $t_0$ and transverse velocities $v_\perp$ that pass reject sampling are then assigned tunnel exit positions $\mathbf{r}_0 = \mathbf{r}_{A/B} + \text{Re}(x_0, 0, 0)$ and longitudinal velocities $v_{0,x}$ where

$$\text{Re}\{x_0(t_0, v_\perp)\} = \frac{E_0 \text{env}(t_0)}{\omega^2} \cos(\omega t_0 + \phi)\left[1 - \sqrt{1 + \gamma^2(t_0, v_\perp)}\right], \tag{4}$$

$$v_{0,x} = \frac{E_0 \text{env}(t_0) \sin(\omega t_0 + \phi)}{\omega}\left[\sqrt{1 + \gamma^2(t_0, v_\perp)} - 1\right], \tag{5}$$

$\mathbf{r}_{A/B}$ is the up-/downfield atomic site (depending on the field sign) and $\text{Re}\{x_0(t_0, v_\perp)\}$ corresponds to the real part of the tunnel exit along the x-axis.

After ionization, electrons propagate semiclassically. Their dynamical positions and momenta are calculated numerically by solving Newton's equation of motion for an electron interacting with the driving laser electric field (1) and two softcore Coulomb forces from the $N_2$ ion, each with $1/2$ fundamental charge at their respective centers:

$$\ddot{\mathbf{r}}(t) = -\mathbf{E}(t) - \nabla V(\mathbf{r}), \tag{6}$$

where the potential $V(\mathbf{r})$ is given by

$$V(\mathbf{r}) = \sum_{j=A,B} -\frac{(1/2)}{\sqrt{[\mathbf{r}(t) - \mathbf{r}_j]^2 + SC}}. \tag{7}$$

In the simulation, $SC = 0.01$ to avoid numerical problems created by the singularities at the atomic centers. Electrons are propagated until the end of the laser pulse $t_1$. During propagation, each electron accumulates a complex phase $\Phi$ derivable from its classical action $S$ [20,24]:

$$\Phi = \int_{t_0}^{t_1} \left(\frac{\mathbf{v}^2}{2} + V(\mathbf{r}) - \mathbf{r} \cdot \nabla V(\mathbf{r})\right) dt - I_p t_0 + \mathbf{v}_0 \cdot (\mathbf{r}_0 - \mathbf{r}_{A/B}) + \Phi_0, \tag{8}$$

where the initial phase $\Phi_0$ accounts for the molecular tilt and is given by [17]

$$\tan \Phi_0 = \tan\left(\frac{v_{z,0} R_0 \sin\theta}{2}\right) \tanh\left(\text{sign}[E_x(t_0)] \frac{R_0 \cos\theta}{2} \sqrt{2I_p + v_{z,0}^2}\right). \tag{9}$$

Once the trajectory calculation is complete, Rydberg electrons are filtered out and final momenta and phases are recorded.

Electron momenta at the detector are determined from the continuum electrons' positions $\mathbf{r}_1$ and momenta $\mathbf{v}_1$ at the end of the laser pulse [25]. Assuming that the electron–molecule interaction can now be approximated as a two-body problem, the asymptotic momentum $\mathbf{v} = (v_x, v_y, v_z)$ is given by

$$\mathbf{v} = v\frac{v(\mathbf{L} \times \boldsymbol{\mathcal{A}}) - \boldsymbol{\mathcal{A}}}{1 + v^2 L^2}, \tag{10}$$

where $\mathbf{L} = \mathbf{r}_1 \times \mathbf{v}_1$ is the angular momentum and $\boldsymbol{\mathcal{A}} = \mathbf{v}_1 \times \mathbf{L} - \mathbf{r}_1/r_1$ is the Runge–Lenz vector, both of which are conserved quantities. The asymptotic momentum magnitude

$v = \sqrt{v_x^2 + v_y^2 + v_z^2}$ comes from solving for the electron's kinetic energy far away from the charged molecule:

$$\frac{v^2}{2} = \frac{v_1^2}{2} - \frac{1}{r_1}. \tag{11}$$

Sample results of the simulation are shown in Figure 2, which plots the 2D asymptotic momentum distribution in the $v_x$-$v_z$ plane. Note that quantum interference is included by attaching a phase to each trajectory, resulting in a complex factor $e^{i\Phi}$ multiplying each trajectory, and accounting for interference between different trajectories that end up with the same final momentum. Additional details about the QTMC simualtions can be found in [14].

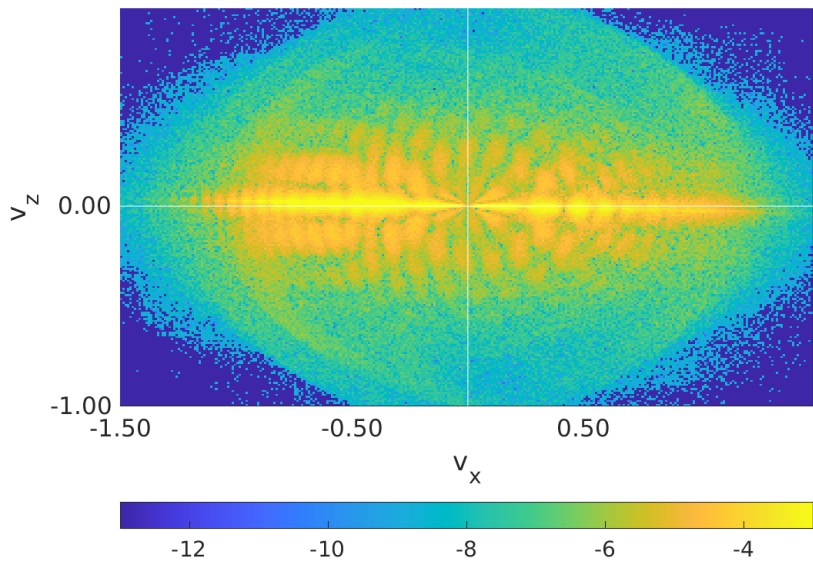

**Figure 2.** Simulated 2D photoelectron momentum distribution (PMD) when the ionization ratio $q = 0.6$ (Equation (12)) and carrier envelope phase $\phi = 0$. In this simulation, a six-cycle, 800-nm laser pulse with peak intensity $I = 1.3 \cdot 10^{14}$ W/cm$^2$ is incident upon a neutral N$_2$ molecule tilted 45 degrees with respect to the polarization direction. When $q = 0.6$, electrons are ionized at the upfield atom four times more often than the downfield atom. The color bar is on a logarithmic scale with arbitrary units.

## 3. Analyzing Momentum Data

To determine the relative number of electrons ionized at either the upfield or downfield locations, we again closely follow the analysis used in reference [14]. First, electron trajectories originating from both atomic sites are calculated. The relative number of up- and downfield electrons used in the analysis is determined by the ionization ratio

$$q = \frac{\# \text{ up} - \# \text{ down}}{\# \text{ up} + \# \text{ down}}, \tag{12}$$

which is sampled within the range $-1$ (all downfield) to $+1$ (all upfield). For each set of trajectories, a 2D photoelectron momentum distribution $w(i,j)$ is generated, where $i$ and $j$ index over bins of $v_x$ and $v_z$, respectively. These momentum distributions are compared to that of experiment [26] by calculating the average offset momentum $a$ for each distribution, where

$$a = \frac{\sum_{i=1}^{m} \text{sign}[v_x(i)] v_{z,\text{mean}}[v_x(i)]}{m}, \tag{13}$$

and

$$v_{z,\text{mean}}[v_x(i)] = \frac{\sum_{j=1}^n w(i,j)v_z(j)}{\sum_{j=1}^n w(i,j)}. \tag{14}$$

In Figure 3, offset momentum $a$ is plotted versus ionization ratio $q$ for various CEP values and compared to the offset momentum calculated from the experimental data. It appears that changing the CEP creates a slight variation in the offset momentum for different $q$ values. However, these ionization ratios for the different CEP all correspond physically to ionizing roughly two downfield electrons for every one upfield.

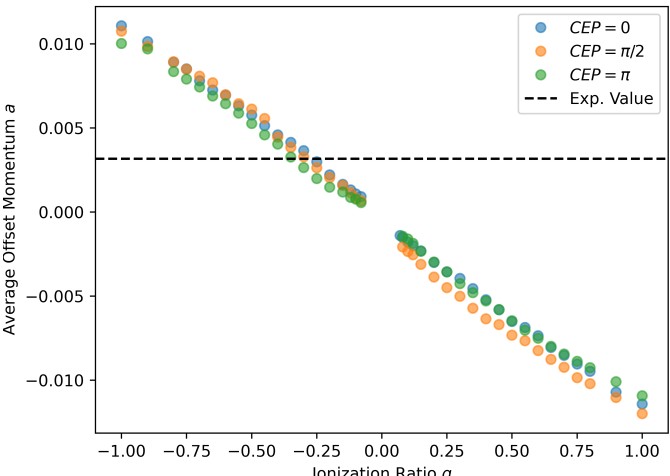

**Figure 3.** Ionization ratio $q$ (Equation (12)) vs. average offset momentum $a$ (Equation (13)) for various values of carrier envelope phase. The experimental offset momentum determined in reference [14] is indicated by the dashed line. It appears that regardless of the CEP, the ratio of downfield to upfield ionization remains roughly 2:1.

## 4. Conclusions

We have simulated the ionization of $N_2$ in a strong electric field through the use of QTMC techniques. By comparing experimental results [26] to simulated electron momentum distributions with various upfield and downfield contributions, we confirm the 2:1 downfield-to-upfield ionization ratio found in [14] regardless of the laser field's carrier envelope phase. Thus, determining the ionization site through this technique does not require experimental CEP stabilization or simulated averaging over CEP values, limiting possible sources of uncertainty.

Importantly, our results support the paradigm of non-adiabatic strong field molecular ionization depending mostly on the Keldysh parameter, $\gamma$, which itself depends only on the laser intensity, frequency, and ionization potential. This view is supported by prior analytical calculations in a static electric field, corresponding to $\gamma \ll 1$, which find that all tunneling is downfield in this fully adiabatic limit [27]. Experimental studies of the strong field ionization of neutral diatomic molecules using longer-wavelength mid-IR pulses, combined with the ionization site extraction procedure proposed in [14], could further test the robustness of this paradigm.

**Author Contributions:** Conceptualization, A.S.L.; simulation and data analysis, A.S. and H.P.; writing—original draft preparation, A.S., H.P. and A.S.L.; writing—review and editing, N/A; supervision, A.S.L.; project administration, A.S., H.P. and A.S.L.; funding acquisition, A.S.L. All authors have read and agreed to the published version of the manuscript.

**Funding:** This research was funded by the U.S. Department of Energy, Office of Basic Energy Sciences, Atomic, Molecular and Optical Sciences Program, under Award No. DE-SC0022093.

**Data Availability Statement:** Data can be provided upon reasonable request.

**Acknowledgments:** We acknowledge prior code development by Lisa Ortmann, which made this work possible.

**Conflicts of Interest:** The authors declare no conflict of interest.

**Abbreviations**

The following abbreviations are used in this manuscript:

| | |
|---|---|
| MDPI | Multidisciplinary Digital Publishing Institute |
| DOAJ | Directory of Open Access Journals |
| CEP | Carrier envelope phase |
| PMD | Photoelectron momentum distribution |
| QTMC | Quantum Trajectory Monte Carlo |

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
