# Peer review of "Does Carrier Envelope Phase Affect the Ionization Site in a Neutral Diatomic Molecule?"

_atoms, doi:10.3390/atoms11040067_

Round 1

Reviewer 1 Report

This work is well presented and provides good extra information about a previous recent work (PRL 127 213201).

Please address the following questions and suggestions:

1. The intensity stated in the caption of Fig. 2 is 1.3e13 W/cm^2, which is different from the value in the main text by an order of magnitude.

2. Is the potential used a two-center pure Coulomb potential? Or some kind of soft core potential? 

3. Is focal averaging included in the simulations? I believe that is crucial for  comparing with experiments. 

4. It would make the paper more illustrative if several more momentum distributions at different peak intensities are presented. Is the conclusion about the 2:1 ratio sensitive to the peak intensity? 

5. For clarity, please show the expression of the phase S.

Reviewer 2 Report

In this work, the authors consider the strong-field ionization of a neutral diatomic molecule and perform quantum trajectory Monte Carlo simulations to gain insight on the whether or not the carrier envelope phase  affects the ionization site.
Their results, based on simulated electron spectra, support the 2:1 downfield-to-upfield ionization ratio previously found by the group director independently of the carrier envelpe phase considered.
The manuscript is well written and provides a valuable contribution to the field.  
I only have a minor point that I would like the authors to tackle. By comparing Fig. 4 of ref.[14] and Fig. 3 of the manuscript, I notice that the simulation values at the ionization ratio q=-1 are not exactly coincident. Independently of the carrier evelope phase, present values are slightly larger. The potential reader will notice this feature as well and it should be desirable to highlight at this point the differences between the former and the present study if this were the case.    
Having resolved this issue the manuscript can be published in Atoms.
